# COVID-19-Vaccination-Induced Myocarditis in Teenagers: Case Series with Further Follow-Up

**DOI:** 10.3390/ijerph19063456

**Published:** 2022-03-15

**Authors:** Mateusz Puchalski, Halszka Kamińska, Marta Bartoszek, Michał Brzewski, Bożena Werner

**Affiliations:** 1Department of Pediatric Cardiology and General Pediatrics, Medical University of Warsaw, 02-091 Warsaw, Poland; mateusz.puchalski@uckwum.pl (M.P.); hkaminska@wum.edu.pl (H.K.); 2Department of Pediatric Radiology, Medical University of Warsaw, 02-091 Warsaw, Poland; marta.bartoszek@uckwum.pl (M.B.); michal.brzewski@wum.edu.pl (M.B.)

**Keywords:** COVID-19, vaccine, myocarditis, teenagers, Cardiology, Paediatrics, cardiac magnetic resonance

## Abstract

Presently, the whole globe is struggling the tough challenge of the COVID-19 pandemic. Vaccination remains the most effective and safe COVID-19 weapon for adults and in the paediatric population. Aside from possible mild and moderate post-vaccination side effects, more severe side effects may occur. We retrospectively analysed a group of 5 teenagers aged from 15 to 17 years with obesity/overweight (BMI ranging from 24.8 to 30) who presented typical myocarditis symptoms following the first or second dose (3 and 2 patients, respectively) of the COVID-19 vaccine. In the whole study group, a significant increase in troponin serum concentration was observed (1674–37,279.6 ng/L) with a further quick reduction within 3–4 days. In all patients, ST segments elevation or depression with repolarisation time abnormalities in electrocardiography were noticed. Chest X-ray results were within normal limits. Echocardiography showed normal left ventricular diameter (47–56.2 mm) with ejection fraction between 61–72%. All patients were diagnosed with myocarditis based on cardiac magnetic resonance (CMR) imaging. During further hospitalisation, swift clinical improvement was notable. Follow-up in the whole study group was obtained after 106–134 days from initial CMR, revealing no myocarditis symptoms, proper troponin level, and no ECG or echocardiographic abnormalities. At the same time, persistent myocardium injury features were detected in the whole study group, including ongoing myocarditis. COVID-19-vaccine-induced myocarditis seems to be a mild disease with fast clinical recovery, but the complete resolution of the inflammatory process may last over 3 months. Further follow-up and investigation for assessing subsequent implications and long-term COVID-19-vaccine-induced myocarditis is required.

## 1. Introduction

The worldwide spread of the coronavirus (COVID-19) pandemic has caused over 5 million deaths across the globe [1]. For the time being, in European countries, two messenger RNA (mRNA) vaccines (Pfizer-BioNTech and Moderna) are approved to reduce the risk, severity, and transmission of the COVID-19 disease in the 12–18 years age group [2]. Beyond mild and moderate post-vaccination side effects, more severe side effects, such as myocarditis, have also been noted [3]. According to the Vaccine Adverse Event Report System (VAERS), 691 cases of COVID-19 vaccine side effects, diagnosed as myocarditis in the aforementioned population, have been reported to date [4]. In our study, we present the clinical characteristics of 5 teenagers hospitalized in the Paediatric Cardiology Department of University Hospital due to acute myocarditis following Pfizer-BioNTech vaccination. Our studied group is one of the first and currently the biggest population with further long-term follow-up with particular emphasis on CMR imaging evaluation in the whole study group.

## 2. Materials and Methods

Our single-centre retrospective study comprised five teenagers hospitalized in the Paediatric Cardiology and General Paediatrics Department from July to August 2021 who were assessed by signs and clinical symptoms, laboratory data of cardiological biomarkers (troponin serum concentration and NT-proBNP), electrocardiography, echocardiography, and cardiac magnetic resonance (CMR). Follow-ups were carried out 106–134 days after initial CMR.

## 3. Results

All patients were white male teenagers (age range 15–17 years) with no underlying medical conditions on admission. Four patients were obese, and one was overweight, according to the World Health Organization’s body mass index [5]. None of them had a positive medical history for previous COVID-19 infection. All patients received the Pfizer-BioNTech mRNA vaccine, 3 of them developed symptoms after the first dose and 2 developed symptoms after the second dose. All presented with symptoms 2–8 days after immunization, with sudden onset sharp and retrosternal chest pain radiating to the left shoulder in patient 5. All but one (patient 4) had body temperature rise, one (patient 2) reported diarrhoea and shoulder pain at the injection spot, and one (patient 4) was affected with dry cough. Symptom duration ranged from 3 to 6 days (Table 1).

On admission, all patients were in a good condition with stable vitals. Each of them was tested for acute respiratory syndrome coronavirus 2 (SARS-CoV-2) by PCR swab test with negative results in the whole group. Other viral and bacterial serologies testing included the most common respiratory tract, neurological, and digestive system pathogens, and were all negative. Blood tests revealed significant increases in highly sensitive troponin I serum concentration in all patients with further rapid, gradual normalization. Furthermore, moderate rise of N-terminal pro-B type natriuretic peptide (NT-proBNP), and C-reactive protein (CRP) plasma concentration was noticed in patients 1 and 4 and patients 1, 2, 3, and 5, respectively, with normalization above mentioned parameters before hospital discharge (Table 1).

Electrocardiogram (ECG) pattern varied, but in all cases, characteristic features for acute myocardium injury, including ST segments elevation or depression, and repolarisation time abnormalities were observed (Table 2). Chest X-rays in all patients revealed no abnormalities. In all patients, transthoracic echocardiography (TTE) showed normal size and function of the left ventricle to the left ventricular internal diameter in diastole (LVIDd) and ejection fraction (EF) within normal limits.

All patients underwent cardiovascular magnetic resonance imaging (CMR) between 6 and 37 days after vaccine immunisation to confirm and evaluate extend of nonischemic myocardium injury and oedema according to updated 2018 Lake Luise Myocarditis Criteria [6]. The obtained CMR images revealed hyperintense signal of oedema partly overlapping with late gadolinium enhancement (LGE) in particular LV segments and thus fulfilling Lake Louise Criteria for myocarditis in the whole group (Table 2). In all patients left and right ventricular function was within normal limits and no pericardial involvement was detected in either the ECHO or the CMR imaging. Considering good general condition with stable vitals, none of the patients qualified for endomyocardial biopsy.

During further hospitalization, all patients were treated with angiotensin-converting enzyme inhibitors (ACEI), and were discharged in stable clinical condition, with no alarming symptoms after 10–16 days of hospital observation and recommendation of a further 3-month follow-up.

Three months after hospital discharge (with the exception of patient 5 with follow-up appointment postponed for one month due to moderate infectious symptoms) renewed evaluation of general condition and additional tests were performed, revealing no symptoms similar to admission time in all except one (patient 4, who reported single, sharp episodes of chest pain lasting for few seconds), normal highly sensitive troponin I serum concentration, regular pattern of ECG in all but one (patient 3, preserved flat T wave in leads V5 and V6), no arrhythmias in 24 h Holter–ECG monitoring, and no abnormalities in TTE. All the patients at this point underwent control CMR (106–134 days after initial CMR). In contrast to other tests, in all patients, CMR imaging showed persistent features of myocardial injury: active inflammatory process (T2-weighted images of myocardium oedema coexisting with LGE in T1-weighted mapping patients 1 and 4) or persistent LGE without oedema. In all patients, ACEI treatment was continued and further treatment was planned (Table 3).

## 4. Discussion

Vaccination against COVID-19 undoubtedly remains the most effective and proved form of COVID-19 elimination in all population groups [7,8]. Moderna and Pfizer-BioNtech mRNA vaccines are distinguished by their high efficacy and safety in children [2,9]. It should be noted that COVID-19 immunisation carries the possibility of mild–moderate side effects (e.g., pain in injection site, fever, malaise occurring and resolving within 1–2 days). In rare cases, more severe adverse effects, even including life-threatening ones, such as myocarditis, may occur [3,10]. It is worth mentioning that post-vaccination myocarditis has been also widely reported after many other vaccines, including influenza, hepatitis B virus, and tetanus [11,12]. The strongest case–effect relation between immunisation and myocardium injury were proven for smallpox vaccination. Endomyocardial biopsy with further pathomorphology study in those cases showed immunological etiology rather than direct myocardium injury [13,14]. As can be seen from the available literature, COVID-19 infection may also provoke subsequent immoderate host immune response [15], so it is possible that, in some rare cases, the same mechanism may be triggered by vaccine administration. The direct mechanism of myocardium injury following the COVID-19 vaccine remains unclear and further investigations are needed [16].

In our study, with a homogeneous teenager group, four patients were described as obese, and one as overweight, according to the WHO body mass index. This may be considered an interesting fact in the light of the recent review by Bortolini et al., according to which, physical inactivity and obesity are factors decreasing the vaccine’s efficacy and immune response [17]. However, obesity itself is a known risk factor of hyperimmune reactions, which may be at least partially responsible for myocarditis in our group. Additionally, quick general condition and symptoms improvement correlated with rapid and prominent decrease in troponin I serum concentration was observed (within 3–4 days after peak, Table 1), and ECG normalization was observed within the whole group.

Similar patterns have been presented in the current literature. In most reported cases, myocarditis after COVID-19 immunisation affects male adolescents or young adults between 12 and 24 years, most commonly after the second dose of the mRNA-type vaccine [18]. The majority of patients presented typical myocarditis symptoms—chest pain, shortness of breath and, fever—within 2–4 days after administration and also with swift improvement of symptoms and general condition. Most patients did not require any advanced therapy or additional procedures [18,19,20,21].

In CMR, the location of myocardial lesions during the acute period (both LGE and oedema) was, in our group, similar to patterns reported so far in the literature (basal and mid-lateral segments of left ventricle) [22,23,24,25]. Despite of significant difference of peak troponin between patients 1, 2, and 3 and patients 4 and 5, there was no correlation between troponin level and CMR findings. The extend of myocarditis in all patients, estimated on the basis of myocarditis involved left ventricle segments, was comparable in all patients. Moreover, elevated troponin concentration, before the outbreak of the COVID-19 pandemic, was recognised as a poor prognostic marker in adult patients with acute coronary syndrome, as well as in children with acute fulminant myocarditis [26]. Contrarily, according to a retrospective by Matsubara et al., longitudinal cohort study troponin concentration seems to not be correlated with worse clinical implications in children with paediatric inflammatory multisystem syndrome temporary, associated with SARS-CoV-2 (PIMS-TS) [27]. In view of the above, and the supposed immune mechanism of PIMS-TS and COVID-19-vaccine-related myocarditis, the troponin level appears to not reflect the CMR findings in wither of the abovementioned groups. This issue still requires further analysis.

Interestingly, the mild and transient clinical course of the disease was not illustrated by the CMR results obtained after 3 months. In all our patients, persistent abnormalities were found, including an active inflammatory process in two of them. To our knowledge, this is one of the first publications highlighting this problem and, so far, the present study is based on the largest number of cases.

In August 2021, Supriya et al. described persistent CMR abnormalities (LGE without oedema) in 2 patients with myocarditis after COVID-19 vaccine in whom CMR was performed after 66 and 71 days from initial CMR [28].

In our group, the active inflammatory process lasted longer. In two patients, after three months, CMR showed persistent oedema with LGE, and in only three LGE.

On the other hand, the most current data from the literature suggest that children with acute myocarditis not related to the vaccination also show CMR abnormalities (mostly LGE) months after full clinical recovery. Seven months after the initial CMR, approximately one third of children still present CMR abnormalities suggestive of active inflammatory processes, and only one third can be described as exhibiting total normalization [29]. Moreover, about half of the patients after myocarditis recovery may present myocardium scarring (consistent with LGE in CMR), which could transform into the trigger spots for even life-threatening rhythm disturbances [30].

In view of the above, our study, with special emphasis on CMR imaging follow-up, symbolises one step forward in obtaining insights into the course and recovery pattern of COVID-19-vaccine-induced myocarditis. Both further studies and longer follow-up of patients should be considered to assess ensuing implications of COVID-19-vaccine-connected myocardium injury.

## 5. Conclusions

COVID-19-vaccine-induced myocarditis seems to be a mild disease with a fast clinical recovery.

The complete resolution of the inflammatory process may last longer.

Based on CMR findings in the study group, the myocardium injury may persist longer than three months from the initial CMR.

Longer follow-up and further investigation for assessing subsequent implications and long-term COVID-19-vaccine-induced myocarditis are required.

## Figures and Tables

**Table 1 ijerph-19-03456-t001:** General information, laboratory tests, treatment, length of hospital stays.

	Case 1	Case 2	Case 3	Case 4	Case 5
Sex	M	M	M	M	M
Age	15	17	17	17	17
BMI kg/m^2^ (WHO centiles)	27 (95), obesity	25.8 (95), obesity	30 (97), obesity	30 (99), obesity	24.8 (89), overweight
Vaccine name	Pfizer-BioNTech	Pfizer-BioNTech	Pfizer-BioNTech	Pfizer-BioNTech	Pfizer-BioNTech
Dose number	2	2	1	1	1
Time (days) from vaccine to symptoms	2	2	2	3	23
Symptom description	Central, non-radiating chest pain, fever (37.8),	Central, non-radiating chest pain, fever (38.5), shoulder pain at the injection site, diarrhoea	Left-sided, non-radiating chest pain, fever (39), dry cough,	Left-sided, non-radiating chest pain	Central, radiating to the left shoulder chest pain, sore throat, fever (39)
Symptom duration (days)	3	3	4	4	6
Laboratory tests:					
COVID-19 PCR test at presentation	Negative	Negative	Negative	Negative	Negative
COVID-19 antibodies					
IgA and IgM class	Negative	Negative	Positive	Positive	Negative
IgG class	Positive	Positive	Positive	Positive	Positive
Troponin concentration					

Peak	14,218.4	37,279.6	10,450.3	1895.2	1674
Next troponin level (days after peak)	102 (4)	354.1 (3)	327.0 (3)	325 (3)	360.0 (3)
Discharge (days after peak)	12.4 (9)	20.9 (13)	18.5 (14)	8.5 (12)	1.5 (11)
Others:					
NT-proBNP	269	27	50	259	391
CRP	3.0	3.9	3.9	0.7	40
White blood cell count	6.8	10.02	7.78	8.02	9.95
Viral and bacterial serological tests					
Respiratory tract panel ^1^	Negative	Negative	Negative	Negative	Negative
Neurological panel ^2^	Negative	Negative	Negative	Negative	Negative
Rotavirus and adenovirus	Not obtained	Negative	Negative	Negative	Negative
Treatment	Lisinopril	Enalapril	Ramipril	Ramipril	Ramipril
Length of hospital stay (days)	10	13	16	12	11

NT-proBNP, pq/mL, (N < 125); Troponin, ng/L (N < 19.0); CRP mg/dL, (N: 0.0–1.0); White blood cell count (N: 4–10 × 10^3^/uL). ^1^ Respiratory tract panel—nasopharyngeal swab; PCR testing—adenovirus, mycoplasma pneumoniae, coronaviruses (HKU1, NL63, OC43, 229E), human metapneumovirus, human rhino/enterovirus, influenzas viruses (A, A/H1, A/H1-2009, A/H3, B, MERS), parainfluenza viruses (1, 2, 3, 4), RSV, SARS-CoV-2, Bordetella pertussis, Bordetella parapertusis, Chlamydophila pneumoniae. ^2^ Neurological panel, blood sample PCR testing—CMV, EBV, HSV1, HSV2, HSV6, HHV6, HHV7, Enterovirus, Adenovirus, Parechovirus, Parvovirus, ParvovirusB19.

**Table 2 ijerph-19-03456-t002:** Imaging tests.

	Case 1	Case 2	Case 3	Case 4	Case 5
**Imaging Tests**					
**Chest X-ray**	Unchanged	Unchanged	Unchanged	Unchanged	Unchanged
**Electrocardiogram**	ST elevation in I, II, V5–V6,	ST elevation in II, III, aVF, V5–V6	ST elevation in V3–V4	ST elevation in II, III, aVF	ST elevation in V4–V6
	ST depression in V1–V2		Negative T wave in I–III, aVL, aVF, V4–V6,		
**Echocardiography**					
**%EF**	64	72	61	62	68
**Left Ventricular End Diastolic Diameter (mm)**	54.5	47–50	45.9	56.2	55
**Regional wall motion changes and pericardial effusion**	No changes	No changes	No changes	No changes	No changes
**CMR imaging**					
**Time from vaccine to CMR (days)**	8	6	10	11	37
**Myocardial oedema (T2 mapping), segment of LV**	All basals	All basals,	Mid: inferior and inferolateral	All basals	All basals
	Mid: anterolateral, inferolateral	Mid: anterolateral, inferolateral,		Mid: inferior, inferolateral	Mid: inferior, inferolateral
		inferoapical: lateral and inferior		Apical: inferior, lateral	
**Nonischemic Myocardial Injury (LGE, T1-weight mapping)**					
**Subepicardial and intraventricular LGE in segments**	Basal: anterolateral, inferolateral, inferior	All basals	No changes	No changes	All basals,
		Mid: inferolateral, inferiors			Mid: inferolateral
		Apical: inferior and lateral			
**Only subepicardial LGE in segments**	Mid: anterolateral, inferolateral	All basals	Basal: inferior and inferolateral	All basals	All basals
		Mid: anterolateral		Mid: inferior, inferolateral	Mid: anterolateral
				Apical: inferior and lateral	

Troponin, ng/L (N < 19.0).

**Table 3 ijerph-19-03456-t003:** Follow-up 106–134 days from initial CMR.

**Follow-Up**					
	**Case 1**	**Case 2**	**Case 3**	**Case 4**	**Case 5**
**Follow-Up**					
**Troponin, ng/L (<19.0)**	4.7	1.5	<1.5	2.3	Not obtained
**ECG findings**	No changes	No changes	Flat T wave in V5–V6	No changes	No changes
**Echocardiography**					
**%Ejection Fraction**	61	66	77	66	77
**Regional wall motion changes**	No changes	No changes	No changes	No changes	No changes
**Pericardial effusion**	No changes	No changes	No changes	No changes	No changes
**CMR imaging**					
**Time from initial CMR (days)**	106	119	112	112	134
**Myocardial oedema (T2 mapping), segment of LV**	All basals	No oedema	No oedema	Basal inferolateral	No oedema
	Mid: anterolateral, inferolateral				
	All inferior				
**Nonischemic Myocardial Injury (Late gadolinium enhancement LGE)**					
**Subepicardial and intraventricular LGE in segments:**	Basal: anterolateral, inferolateral, inferior	All basals,	No changes	No changes	All basals
		Mid: inferolateral,			Mid: inferolateral, inferior
		All inferior			
		Apical lateral			
**Only subepicardial LGE in segments:**	No changes	No changes	Basal: inferior and inferolateral	All basals	No changes
				Mid: inferior, inferolateral	
				Apical: inferior, lateral	

## Data Availability

The data will be made available by the corresponding author, upon reasonable request.

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
