# Peer review of "COVID-19-Vaccination-Induced Myocarditis in Teenagers: Case Series with Further Follow-Up"

_ijerph, 2022, doi:10.3390/ijerph19063456_

Round 1

Reviewer 1 Report

I read with interest the paper by Puchalski et al., describing a case-series of myocarditis after COVID-19 vaccination. The topic is actual and prevalence of myocarditis, pericarditis or both is probably underestimated. The manuscript is quite well written. The methods are adequate for the type of report. I gave minor comments for the authors:

  • It is not clear, for the CMR data, if there was a pericardial involvement in any of the 5 cases report. Please specify.
  • It would be interesting to ECG monitoring for cardiac arrhythmias or subsequent investigation once discharged from the hospital
  • all patients described in these case-series was overweight or obese. Please discuss if the obesity itself may favor a hyperimmune response to vaccine (refer for example to Int J Environ Res Public Health. 2022 Feb 7;19(3):1853)
  • Please report and compare your data to existing extensive dataset on myocarditis after COVID-19 vaccination(refer to Circulation. 2022 Feb;145(5):345-356. JAMA. 2022 Jan 25;327(4):331-340)
  • Cardiac troponin decrease remarkably after some days after hospitalization. Could you further comment these data?
  • Please explain why patients underwent treatment only with ACE-i and not with non-steroidal anti-inflammatory drugs.

Please revise typos.

Author Response

Dear Reviewer,

   At the beginning, we would like to thank you for your interest in our manuscript. We appreciate the time and effort you have dedicated to provide your valuable feedback on our manuscript. Here, we would like to present, poin-by-point, response to your comments and concerns. We thoroughly revised the manuscript text according to yours suggestions. Hereunder, we include a response to each question and for all  concerns.

  1. It is not clear, for the CMR data, if there was a pericardial involvement in any of the 5 cases I Please specify.

Author response :Thank you for pointing this out. There was no pericardial involvement in any of our patients in cardiac magnetic resonance imaging– we have put this information in the text.

  1. It would be interesting to ECG monitoring for cardiac arrhythmias or subsequent investigation once discharged from the hospital

Author response : All the patients underwent ECG and 24-hours-Holter-ECG monitoring – we have specified it in our manuscript

  1. All patients described in these case-series was overweight or obese. Please discuss if the obesity itself may favor a hyperimmune response to vaccine (refer for example to Int J Environ Res Public Health. 2022 Feb 7;19(3):1853)

Author response : Thank you for bringing our attention to this interesting and recent paper – unfortunatelly it had been published after our manuscript was sent for review. Presently we have adapted our text as follows:

In our study, with homogeneous teenagers group 4 patients were described as obese and one as overweighed according to body-mass WHO index. This may be considered an interesting fact in the light of the recent review by Bortolini et al according to which physical inactivity and obesity are the factors decreasing vaccine’s efficacy and immune response. However, obesity itself is a known risk factor of hyperimmune reactions which may be at least partially responsible for myocarditis in our group. 

  1. Please report and compare your data to existing extensive dataset on myocarditis after COVID-19 vaccination(refer to Circulation. 2022 Feb;145(5):345-356. JAMA. 2022 Jan 25;327(4):331-340)

Author response : The data collected and analyzed in those recent and extensive publications (both of them released after drafting our manuscript) we presently mention in our manuscript. Still, we put forward the main strength of our small population – the CMR follow-up which was not addressed so far in any publication – including those mentioned above.

  1. Cardiac troponin decrease remarkably after some days after hospitalization. Could you further comment these data?

Author response : We have placed the extra details illustrating the troponin decrease during hospitalization in Table 1. This rapid improvement is commented in other publications as well – we do mention it in the discussion.

  1. Please explain why patients underwent treatment only with ACE-i and not with non-steroidal anti-inflammatory drugs.

Author response : According to current recommendations (American Heart Association – Diagnosis and Management of Myocarditis in Children, 2021) in acute myocarditis only medical treatment  for heart failure, atrial or ventricular arrhythmias is advised. According to above mentioned guidelines, in our Clinical Center we do not use non-steroidal anti-inflammatory drugs in myocarditis routinely.

We hope that changes implemented in our revised manuscript will satisfy you.

Sincerely yours,

Prof. Bożena Werner MD PhD

Reviewer 2 Report

Several recent papers and letters reported case series of myocarditis in children and teeagers after COVID-19 vaccination.  The experience reported by Puchalski and coworkers is small, focusing on a single center study population.

The reported data are original given that this represents a clinical problem with implications in preventive medicine.

The clinical approach in the diagnosis has been adequate and correct, including several lab diagnostic test , to esclude the etiological role of other infective agents. The cardio CMR study has been carried out and reported according to LLC criteria. The discussion otherwise reported other experiences in a less articulate and rational way ; the conclusions are synthetic but effective.

Author Response

Dear Reviewer,

   At the beginning, we would like to thank you for your interest in our manuscript. We appreciate the time and effort you have dedicated to provide your valuable feedback on our manuscript. 

Thank you for your remarks – we have made an effort to improve the discussion and we hope it will earn your appreciation. At the same time, we would like to emphasize that despite of constanly growing numbers of case series of myocarditis in children after COVID-19 vaccination, our manuscript, best of our knowledge, is the first with cardiac magnetic resonance follow-up.

We hope that changes implemented in our revised manuscript will satisfy you.

Sincerely yours,

Prof. Bożena Werner MD PhD

Reviewer 3 Report

This is a case series about myocarditis after covid vaccination in teenager. It only included 5 cases with MRI findings. There were already several reports about myocarditis after covid vaccination including MRI findings.  However, this study included follow-up findings of MRI in the long term after myocarditis development., which might be valuable. However, there were several issues to be addressed.

# How about the right ventricular function in cardiac MRI?

# The value of CRP for patient 5 was 40. Was it true?

# The value of peak troponin was significantly different between patient 1/2/3 and patient 4/5. How about any differences between these two groups about MRI findings?

Author Response

Dear Reviewer,

 At the beginning, we would like to thank you for your interest in our manuscript. We appreciate the time and effort you have dedicated to provide your valuable feedback on our manuscript. Here, we would like to present, poin-by-point, response to your comments and concerns. We thoroughly revised the manuscript text according to yours suggestions.

Hereunder, we include a response to each your question and concern.

We hope that changes implemented in our revised manuscript will satisfy you.

  1. How about the right ventricular function in cardiac MRI?

Authors’ response: The function of both vemntricles in MRI imaging was within normal limits – we have put this information in the manuscript text. 

  1. The value of CRP for patient 5 was 40. Was it true?

Authors’ response: The value of CRP for patient 5 was 40 mg/dl indeed. However during further hospitalization quick decreasing with CRP level normalization before hospital discharge was also observed. 

  1. The value of peak troponin was significantly different between patient 1/2/3 and patient 4/5. How about any differences between these two groups about MRI findings?

Authors’ response: – Despite of significant difference of peak troponin between patients number 1/2/3/ and patients 4/5, there was no correlation between troponin level and MRI findings. The extend of myocarditis in all patients, estimated on the basis of myocarditis involved left ventricle segments, was comparable in all patients. Moreover, elevated troponin concentration, before COVID-19 pandemia outbreak, was recognised as  a poor prognostic marker in adult patients with acute coronary syndrome as well as in children with acute, fulminant myocarditis. Contrary, according to Matsubara et al. retrospective, longitudinal cohort study troponin concentration seems not to be correlated with worse clinical implications in children with Pediatric Inflammatory Multisystem Syndrome Temporary Associated with SARS-CoV-2 (PIMS-TS)*. In view of the above and supposed immune mechanism of PIMS-TS and COVID-19 vaccine-related myocarditis the troponin level appears to not reflect the MRI findings in both  above mentioned patients groups. This issue still requires further profound analysis. – we have  emphasized it in the text.

* -Matsubara, D.; Chang, J.; Kauffman, H.L.; Wang, Y.; Nadaraj, S.; Patel, C.; Paridon, S.M.; Fogel, M.A.; Quartermain, M.D.; Banerjee, A. Longitudinal Assessment of Cardiac Outcomes of Multisystem Inflammatory Syndrome in Children Associated With COVID-19 Infections. J. Am. Heart. Assoc., 2022, 11, DOI: 10.1161/JAHA.121.023251. 

 Sincerely yours,

Prof. Bożena Werner MD PhD

Round 2

Reviewer 3 Report

Revised manuscript was finely corrected.